

# Reduction in black carbon light absorption due to multi-pollutant emission control during APEC China 2014

Yuxuan Zhang[1,2], Xin Li[1], Meng Li[1,2], Yixuan Zheng[1], Guannan Geng[1], Chaopeng Hong[1], Haiyan Li[3], Dan Tong[1], Xin Zhang[1], Yafang Cheng[4,2], Hang Su[4,2], Kebin He[3], and Qiang Zhang[1]

[1] Department of Earth System Science, Tsinghua University, Beijing 100084, China
[2] Multiphase Chemistry Department, Max Planck Institute for Chemistry, Mainz 55020, Germany
[3] State Key Joint Laboratory of Environment Simulation and Pollution Control, School of Environment, Tsinghua University, Beijing 100084, China
[4] Institute for Environmental and Climate Research, Jinan University, Guangzhou 510630, China

*Correspondence to*: Qiang Zhang (qiangzhang@tsinghua.edu.cn)

**Abstract.** Reducing black carbon (BC) emissions has been recognized as an efficient way to simultaneously improve air quality and mitigate climate change. However, the benefits of BC emission controls are not well quantified partly due to a lack of understanding of the changes in BC light absorption as a result of emission reductions. In this work, we discussed the effects of multi-pollutant emission reductions on the BC light absorption based on a field campaign study conducted before, during and after the 2014 APEC (Asia-Pacific Economic Cooperation) meeting in Beijing, China. When emission restrictions were in place during APEC, we found that the reduction in the light absorption of BC-containing particles was driven by both the decrease in BC mass concentration and the weakened light-absorption capability of BC. Compared with that before and after APEC, the daytime light absorption of BC-containing particles during APEC reduced by ~56%, of which ~48% was contributed by the decrease in BC mass concentration and the remain ~8% was contributed by a weakening of light-absorption capability for BC. Based on single particle soot photometer (SP2) measurement and Mie calculation, we estimated that the light-absorption capability of BC-containing particles with ~80-200 nm refractory BC (rBC) cores at daytime during APEC was reduced by ~6-15% and ~10-20% compared with that before and after APEC, respectively. The decrease in BC light-absorption capability could be attributed to less coating materials on BC surfaces as a result of a decreased chemical production of secondary aerosols. Compared with that before and after APEC, the mass ratio between the coating materials and rBC core (~80-200 nm) during APEC decreased by ~10-30% and ~31-53%, respectively, due to reductions in coating precursor emissions, e.g., $SO_2$ and $NO_2$. The results revealed the benefits of emission control on BC light absorption by simultaneously reducing the mass concentration and light-absorption capability of BC, implying that synergetic reduction in multiple-pollutant emission could benefit both air quality and climate.

## 1 Introduction

Black carbon (BC) has drawn considerable attention due to its key role in climate and the atmospheric environment (Bond



and Sun, 2005; Jacobson et al., 2002, 2010). Because BC is the most efficient light-absorbing component in ambient aerosols
(Bond and Bergstrom, 2006; Ramanathan and Carmichael, 2008), reduction measures targeting BC emissions have been
recognized as a viable way to mitigate global warming (Shindell et al. 2012; Jacobson et al., 2010) and improve air quality in
polluted regions (Ding et al., 2016; Wang et al., 2018). The benefits of BC emission reduction are mainly driven by more
solar radiation reaching the surface due to the reduction in BC light absorption in the atmosphere.

6       The light absorption of ambient BC-containing particles can be reduced by decreasing the BC mass concentration,

weakening the BC light-absorption capability or implementing both strategies. As primary aerosols, the mass concentration
of BC particles generally decreases with emission reduction. When emission control measures were implemented, the mass
concentration of the BC present in the atmosphere was proven to decrease (Han et al., 2015; Huang et al., 2010; Xu et al.,
2015; Zhang et al., 2016a). In terms of the influence of emission reduction on the characteristics of BC aerosols, previous
studies usually highlighted the decrease in BC mass concentration (Han et al., 2015; Wang et al., 2018; Zhang et al., 2016a).
However, few studies considered the change in light-absorption capability of BC-containing particles due to emission
reduction.
The light-absorption capability of ambient BC-containing particles is closely associated with their aging degree
(Jacobson et al., 2001; Liu et al. 2017; Moffet et al., 2009; Peng et al., 2016; Zhang et al. 2016c, 2018), i.e., the degree to
which BC is internally mixed with other species (e.g., sulfate and nitrate) (Oshima et al., 2009). When fresh BC is emitted
from incomplete combustion, most of BC particles are externally mixed with other aerosol components (e.g., primary
organic aerosol). These fresh BC particles exist as almost bare particles with few other species condensed on their surfaces
and are named externally mixed BC particles (Jacobson et al., 2001; Chung et al. 2005). During atmospheric transport, fresh
BC particles undergo aging, in which internally mixed BC particles form when other aerosol components coat the bare BC
surface (Cheng et al., 2006; Bond and Bergstrom, 2006; Peng et al., 2016; Zhang et al., 2018). The internally mixed BC
particles generally have a shell-and-core morphology, with the coating materials and BC as the shell and core, respectively.
This shell-and-core morphology endows BC particles with a higher light-absorption capability because the coating materials
act as a lens to focus more photons on BC (lensing effect, Lack and Cappa 2010). Compared with externally mixed BC
particles (i.e., bare BC), the light absorption of internally mixed BC particles (i,.e, coated BC) can be enhanced by a factor of
2-3 (Fuller et al., 1999; Jacobson et al., 2001; Schnaiter et al., 2005; Zhang et al., 2016c).
Emission reduction may affect the lensing effect by changing the amount of coating materials for the BC-containing
particles and consequently altering the light-absorption capability of BC. Emission control measures can reduce the
concentrations of not only BC but also co-emitted gaseous pollutants (e.g., volatile organic compounds (VOCs), $SO_2$ and
$NO_x$) present in the atmosphere (Tang et al., 2015; Huang et al., 2015). The reduction in these secondary aerosol precursors
can lower the production of secondary components (e.g. secondary organic matter, sulfate and nitrate) in aerosol particles
(Cheng et al., 2008; Huang et al., 2010; Han et al., 2015). This relationship implies that the interaction between BC and
secondary aerosol components via condensation and coagulation may be impacted by the primary emission reductions of
both BC and co-emitted pollutants (e.g., VOCs, $SO_2$ and $NO_x$), namely, emission control measures may influence BC aging



in the atmosphere. As mentioned above, the aging degree of BC-containing particles exerts a substantial effect on their light-
absorption capability. Less aged BC is expected as emission control measures are implemented to decrease BC light-
absorption capability. However, it is still unclear whether emission control measures can lower the aging degree of BC-
containing particles and thus weaken their light-absorption capability.

5        In this work, we used the 2014 Asia-Pacific Economic Cooperation (APEC) meeting in Beijing, China, as a case study to

investigate the effects of emission control measures on the light absorption of ambient BC-containing particles. This paper
reported the in situ measurements before, during and after APEC and investigated how the concentrations of BC and coating
precursors, the BC aging degree and the BC light-absorption capability were affected by emission reductions. Based on these
results, we quantified the impact of emission reduction during APEC on the light absorption of BC-containing particles and
further discuss the additional effect of emission control measures on BC light absorption due to changes in the coating
materials of ambient BC particles.

## 2 Methods and data

### 2.1 Measurement location and period

The in situ measurement was carried out on the campus of Tsinghua University (40º00'17" N, 116º19'34" E). The
observation site is located in downtown Beijing, approximately 1 km from North 4th Ring Road, which has a high traffic
density. The air quality at this site is considered typical of the Beijing urban environment. More details regarding the
Tsinghua site can be found in Zheng et al., (2015) and Zhang et al. (2018).

18       The measurement period lasted from October 28 to November 21, 2014. A series of aggressive measures were

implemented from November 3 to 12, 2014 in Beijing and the surrounding areas (i.e., Tianjin, Hebei, Shanxi, Shandong,
Henan and Inner Mongolia) to achieve good air quality during the APEC meeting: mandatory restrictions on traffic flow in
Beijing, limited or arrested production from high-emitting factories, suspended construction activities and bans on various
outdoor burning practices (Gao et al., 2017; Huang et al., 2015; Tang et al., 2015; Zhang et al., 2016a; Zhang et al., 2016b).
In this study, we classified the observation period into five subperiods: before APEC (October 28-November 2, 2014), which
served as a reference; during APEC (November 6-12, 2014), which was characterized by the enforcement of emission
control measures; after APEC (November 17-21, 2014), which served as another reference; and two transition periods
(November 3-5 and 13-16, 2014), which were defined considering the lifetime of BC during atmospheric transport and not
discussed in this work.

### 2.2 Instrumentation

A single-particle soot photometer (SP2) instrument (Droplet Measurement Technologies, Boulder, CO, USA) uses a 1064
nm Nd:YAG laser to measure the mass of a refractory BC (rBC) core ($m_{rBC}$) and the scattering cross section ($C_s$) of an



individual BC-containing particle. As a light-absorbing component, a rBC core is gradually heated by the continuous laser
beam and vaporizes at ~4000 K, where detectable incandescent light is emitted (Schwarz et al., 2006; Moteki and Kondo,
2010). The incandescence signal recorded by SP2 was used to determine the $m_{rBC}$ of an individual BC-containing particle.
The mass concentration of rBC was calculated based on the $m_{rBC}$ and sampling flow rate (0.12 lpm (liter per minute)). On the
other hand, we used the scattering signal from the SP2 measurement to retrieve the $C_s$ of an individual BC-containing
particle (including coating materials and rBC core) based on the leading-edge-only (LEO) method developed by Gao et al.
(2007). The validity of the LEO method for ambient BC-containing particles observed in China has been evaluated by Zhang
et al. (2016c). More details on the SP2 technique have been reported elsewhere (Gysel et al., 2011; Pan et al., 2017; Sedlacek
et al., 2012; Zhang et al., 2016c).
The observational data of hourly $PM_{2.5}$, $SO_2$, $NO_2$ and $O_3$ concentrations at the Wanliu station in urban Beijing were
downloaded from the Atmospheric Environment Monitoring Network (http://www.zhb.gov.cn/). The Wanliu station is
approximately 5 km away from the Tsinghua site**.**
**2.3 Data analysis**
**2.3.1 Aging degree of BC-containing particles**
The aging degree of ambient BC-containing particles was retrieved by the SP2 measurements (i.e., the $m_{rBC}$ and the $C_s$ of
BC-containing particles) and Mie calculation. To quantify the aging degree of BC-containing particles, we assumed that a
BC-containing particle was a sphere with a rBC core and a non-refractory coating material (NR-CM) shell (Moteki and
Kondo, 2007; Subramanian et al., 2010; Zhang et al., 2016c). In this study, the diameter of the rBC core ($D_c$) and the whole
particle diameter including the core and shell ($D_p$) were calculated to retrieve the aging degree of BC-containing particles. $D_c$
was calculated from $m_{rBC}$ and the density of the rBC core ($\rho_c$, here, a prescribed value of 1.8 g cm$^{-3}$) (Cappa et al., 2012; Pan
et al., 2017; Laborde et al., 2013). $D_p$ was determined via the Mie calculation and was related to the $D_c$, the $C_s$ of the BC-
containing particle, and the refractive indices of NR-CM ($RI_{NR-CM}$) and rBC core ($RI_c$). The values of $RI_{NR-CM}$ and $RI_c$ used in
this study were 1.5-0i and 2.26-1.26i, respectively (Cappa et al., 2012; Taylor et al., 2015). More details regarding the
calculation of $D_p$ and $D_c$ for ambient BC-containing particles observed in Tsinghua site can be found in Zhang et al. (2018).
In this study, the aging degree of a BC-containing particle was characterized by the mass ratio between NR-CM and
rBC ($m_{NR-CM}/m_{rBC}$) and was calculated by Eq. (1):
$$\frac{m_{NR-CM}}{m_{rBC}} = \frac{\frac{1}{6} \times \pi \times (D_P^3 - D_c^3) \times \rho_{NR-CM}}{m_{rBC}}$$    (1)
where $m_{NR-CM}$ is the mass of the non-refractory coating materials; $\rho_{NR-CM}$ is the density of the non-refractory coating
materials, with a prescribed value of 1.4 g cm$^{-3}$ in this study based on the composition of submicron aerosols during APEC
reported by Zhang et al. (2016a) and the densities of the various components (i.e., sulfate, nitrate, ammonium and organic
aerosol) (Cappa et al., 2012).



**2.3.2 Light absorption of BC-containing particles**
In this study, the light-absorption capability of ambient BC-containing particles was characterized by the light absorption
enhancement ($E_{ab}$) of BC from the lensing effect caused by the coating materials. The $E_{ab}$ of BC-containing particles was
retrieved using a shell-and-core model based on Mie theory (Laborde et al., 2013; Metcalf et al., 2013; Schwarz et al., 2008),
calculated by dividing the light absorption cross-section of the whole BC-containing particle ($C_{ab,p}$) by that of the bare rBC
core ($C_{ab,c}$) at a certain wavelength (550 nm in this study), as expressed in Eq. (2):
$$E_{ab} = \frac{C_{ab,p}\ (D_c, D_p, RI_{NR-MC}, RI_c)}{C_{ab,c}(D_c, RI_c)} \qquad (2)$$
where $C_{ab,c}$ and $C_{ab,p}$ were determined from the Mie calculation. $C_{ab,c}$ is related to $D_c$ and $RI_c$. For $C_{ab,p}$, we needed additional
information on the whole particle, i.e., $D_p$ and $RI_{NR-CM}$.
The light absorption coefficient ($\sigma_{ab}$) of BC-containing particles at 550 nm was determined by the light-absorption
capability of BC and the rBC mass concentration ($C_{rBC}$), as shown in Eq. (3):
$$\sigma_{ab} = C_{rBC} \times E_{ab} \times MAC_c \qquad (3)$$
where $MAC_c$ is the mass absorption cross-section (MAC) of rBC cores, which was prescribed a value of 7.5 $m^2\ g^{-1}$ at 550 nm
(Bond and Bergstrom, 2006).
**3 Results**
**3.1 Reduction in the concentrations of BC and coating precursors**
Figure 1a shows the time series of the $PM_{2.5}$ and rBC mass concentrations during the campaign period. Three pollution
episodes on October 28-November 1, November 6-11 and November 17-21 were observed before, during and after APEC,
respectively. In this work, we focused on comparing the BC characteristics among the three pollution episodes. The $PM_{2.5}$
concentration during the pollution episodes before and after APEC were ~127 μg m$^{-3}$ and ~213 μg m$^{-3}$, respectively, which
were larger than that (~66 μg m$^{-3}$) during APEC. The decrease in $PM_{2.5}$ loadings revealed that the air quality was improved
during APEC. Similarly, the rBC mass concentration during APEC was also smaller than that before and after APEC.
However, the decreases in the rBC concentration during APEC by ~27% and ~58%, respectively, compared with that before
and after APEC were smaller than the corresponding decreases in the $PM_{2.5}$ concentrations (~48% and 69%, respectively),
possibly indicating that more secondary aerosols (e.g., sulfate and nitrate) than primary aerosols (e.g., rBC) were reduced
during APEC, which could aid the decrease in coating materials on BC surfaces.
Fig. 2 compares the mass concentrations of both rBC and the coating precursors (i.e., $NO_2$ and $SO_2$) in the pollution
episodes before, during and after APEC. Compared with that before and after APEC, the mass concentration of $NO_2$ during
APEC was decreased by ~34% and ~45%, respectively, while the $SO_2$ concentration was reduced by ~35% and ~67%,
respectively. These results revealed that the emission control measures implemented during APEC were a viable way to





reduce not only the rBC mass concentrations but also the concentrations of secondary aerosol precursors present in the
atmosphere. The emission control-caused reduction in secondary particle precursors (i.e., $NO_2$ and $SO_2$) during APEC could
have reduced the secondary aerosol formation in the atmosphere. Previous studies identified a reduction in the
concentrations of secondary components (e.g., sulfate and nitrate) in aerosols during APEC compared to that before and after
APEC (Zhang et al., 2016a; Han et al., 2015), indicating that the production of BC coating materials might have been
affected by emission controls during APEC.

7       Given the importance of photochemical reactions in BC aging process (Metcalf et al., 2013; Zhang et al., 2014; Peng et
al., 2016), changing the daytime concentrations of rBC and coating precursors might play a more important role in affecting
BC aging than altering the nighttime concentrations. We separated the data sets for the pollution episodes before, during and
after APEC into daytime (07:00-19:00) and nighttime (19:00 to 07:00 of the following day) sets. Fig. 2 shows that while the
emission controls were in place during APEC, a greater reduction in the rBC and $NO_2$ concentrations occurred during the
day than at night. Compared with those before and after APEC, the daytime reductions in the $NO_2$ concentration during
APEC were reduced by as much as ~40% and ~51%, respectively. By contrast, the daytime reduction (~25%) in the $SO_2$
concentration during APEC compared with that before APEC was smaller than that at night, which might be attributable to
the high contribution of regional emissions (e.g., power generation and industrial activities in Hebei Province) to the daytime
$SO_2$ concentration in Beijing (Guo et al., 2014; Tang et al., 2015). Meanwhile, a similar reduction (~67%) in the daytime and
nighttime $SO_2$ concentrations during APEC compared with that after APEC was observed. In summary, the significant
reductions in the daytime levels of rBC and coating precursors during APEC further indicated that the BC aging in the
atmosphere might have been affected by the emission control measures.
**3.2 Reductions in the aging degree of BC**
Figs. 1b and 1c show time series of the number size distribution of rBC cores ($D_c$) and whole BC-containing particles ($D_p$),
respectively. The rBC cores observed before, during and after APEC exhibited similar number size distributions, with a
mode at ~95 nm (Fig. 1b). The similar modes of the rBC cores could have resulted from similar emission sources and similar
atmospheric processes (coagulation and wet removal) for BC particles observed before, during and after APEC. However,
the whole BC-containing particles (including coating materials and rBC core) showed different number size distributions in
the pollution episodes before, during and after APEC (Fig. 1c), indicating different amounts of coating materials on the BC
surface during the three pollution episodes. In the pollution episodes before and after APEC, the particle size of the whole
BC-containing particles exhibited sustained growth from ~180 nm to ~320 and ~400 nm, respectively, which could be
attributed to the gradual condensation of secondary species on the BC surface. However, the continuous size growth of the
whole BC-containing particles was not observed in the pollution episode during APEC, in which the number particle size
distribution with a mode no more than 280 nm (Fig. 1c), significantly smaller than that before (~320 nm) and after APEC
(~400 nm). These results indicated that secondary formation during APEC was insufficient to maintain continuous BC aging.
Fig. 3 compares the mass ratio between the coating materials and rBC cores ($m_{NR-CM}/m_{rBC}$) for BC-containing particles



with size-resolved rBC cores in the pollution episodes before, during and after APEC. The $m_{NR-CM}/m_{rBC}$ ratios of BC-
containing particles before, during and after APEC showed similar correlations with the rBC core size, namely, $m_{NR-CM}/m_{rBC}$
decreased with increasing rBC core size (Fig. 3a). The size-dependent $m_{NR-CM}/m_{rBC}$ ratio of BC-containing particles indicated
that condensational growth was more effective for smaller particles. At a certain size of rBC cores, Fig. 3a shows that $m_{NR-}$
$_{CM}/m_{rBC}$ ratio of ambient BC-containing particles during APEC was significantly smaller than that before and after APEC,
revealing that the emission restrictions during APEC weakened the condensation of other species on the BC surface. For
ambient BC-containing particles with ~80-200 nm rBC cores, the $m_{NR-CM}/m_{rBC}$ ratios observed in the pollution episodes
before, during and after APEC were 4-22, 3-15, and 5-33, respectively.

9        Fig. 3b shows the reductions in $m_{NR-CM}/m_{rBC}$ ratio of BC-containing particles for the pollution episodes during APEC

compared with that before and after APEC, which were also dependent on rBC core size. Smaller rBC cores exhibited
greater reductions in the $m_{NR-CM}/m_{rBC}$ ratio as a result of emission control measurements during APEC, indicating that in
terms of BC aging, it was more sensitive to emission levels for smaller rBC cores. This could be explained by the diffusion-
controlled growth law, i.e., the condensational growth of smaller BC particles was more effective (Metcalf et al., 2013;
Seinfeld and Pandis, 2006), and thus, the effect of emission reduction on BC aging was more significant for smaller rBC
particles. Compared with that before and after APEC, the $m_{NR-CM}/m_{rBC}$ ratio of ambient BC-containing particles with ~80-200
nm rBC cores during APEC was reduced by ~10-30% and ~31-53%, respectively. The relationship between the reduction in
$m_{NR-CM}/m_{rBC}$ ratio of BC-containing particles ($R_{aging}$) during APEC and their rBC core size ($D_c$) followed an exponential
function (Fig. 3b), i.e., $R_{aging} = 9.1+1576.6\exp(-0.055D_c)$ (relative to that before APEC) and $R_{aging} = 30.7+169.2\exp(-$
$0.025D_c)$ (relative to that after APEC).

20        The reduction in $m_{NR-CM}/m_{rBC}$ ratio of BC-containing particles for the pollution episode during APEC relative to that

before and after APEC showed pronounced diurnal cycles (Fig. 4). Compared with that before APEC, the reduction in $m_{NR-}$
$_{CM}/m_{rBC}$ ratio of BC-containing particles with 80-200 nm rBC cores during APEC showed maxima in the afternoon (~14:00-
17:00 LT) (Fig. 4a), consistent with the peak time of the diurnal cycle of $O_3$ concentrations before and during APEC (Fig.
4c). This consistence indicated that the reduction in coating materials on the BC surface during APEC compared to that
before APEC was most likely dominated by a lower photochemical production of secondary species. Fig. 5a1 shows that the
reduction in $m_{NR-CM}/m_{rBC}$ ratio of BC-containing particles during APEC relative to that before APEC increases with the $O_3$
concentration during the day (7:00-19:00 LT), revealing that the effect of emission controls on BC aging is associated with
photochemistry. Moreover, Fig. 4a shows the diurnal cycle of the reduction in $m_{NR-CM}/m_{rBC}$ ratio of BC-containing particles
during APEC compared to that before APEC with minima during rush hour (~6:00-8:00 LT), which can be due to a larger
contribution of primary emissions of fresh BC (namely, bare BC and thin coated BC particles) during rush hour than at other
times for both episodes before and during APEC.

32        However, the reduction in $m_{NR-CM}/m_{rBC}$ ratio of BC-containing particles for the pollution episode during APEC compared

to that after APEC showed a different diurnal cycle, with maxima at ~10:00-12:00 LT and with minima at ~15:00-17:00 LT.
Fig. 4c shows that the daytime $O_3$ concentrations after APEC are significantly smaller than those during APEC, indicating a





weakened contribution from photochemistry after APEC. The increased amount of coating materials of BC observed after
APEC compared to that during APEC was mostly likely attributed to enhanced heterogeneous chemistry during haze
episodes (Xie et al., 2015; Yang et al., 2015; Zheng et al., 2015). Fig. 5a2 shows that the variation in the reduction in $m_{NR\text{-}CM}/m_{rBC}$
ratio of BC-containing particles during APEC compared to that after APEC is poorly correlated with the $O_3$
concentration. The diurnal trend of the reduction in $m_{NR\text{-}CM}/m_{rBC}$ ratio of BC-containing particles during APEC relative to
that after APEC was likely driven by the simultaneous effects of enhanced photochemistry and weakened heterogeneous
chemistry contributions during APEC.

8       As discussed above, the reduction in the aging degree of ambient BC-containing particles during APEC could have been

caused by a decreased chemical production (namely, weakened contributions from photochemical or heterogeneous
reactions) of coating materials on the BC surface. Fig. 5b shows that the reduction in the $m_{NR\text{-}CM}/m_{rBC}$ ratio of BC-containing
particles during APEC relative to that before and after APEC is associated with a decrease of the concentrations of $SO_2$ and
$NO_2$ due to emission reduction. A greater decrease in the concentrations of $SO_2$ and $NO_2$ corresponded to a greater reduction
in the $m_{NR\text{-}CM}/m_{rBC}$ ratio of BC-containing particles during APEC. The reduction in precursor emissions of secondary species
(e.g., $SO_2$ and $NO_2$) could decrease the chemical production, and therefore, lower amounts of coating materials on the BC
surfaces were observed during APEC.

## 3.3 Reduction in the light absorption of BC-containing particles

The reduction in the BC aging degree during APEC should have weakened the light-absorption capability of BC-containing
particles owing to a decrease in the lensing effect caused by less coating materials on the BC surfaces (Fuller et al. 1999;
Lack and Cappa 2010). Fig. 6 compares the $E_{ab}$ of BC-containing particles during the day for the pollution episodes observed
before, during and after APEC. The daytime $E_{ab}$ of BC-containing particles with 80-200 nm rBC cores varied from ~1.5 to
~2.5 during APEC, values that were remarkably lower than that before and after APEC (i.e., $E_{ab}$ of 1.7-3.0 and 1.8-3.2,
respectively, Fig. 6a); these results reflected a weakened light-absorption capability of BC during APEC. The reduction in
the daytime $E_{ab}$ of BC-containing particles ($R_{Eab}$) during APEC compared with that before and after APEC decreased with
the rBC core size ($D_c$), and the relationship followed an exponential function ($R_{Eab} = 6.3+192.9\exp(-0.039D_c)$ (relative to that
before APEC) and $R_{Eab} = 9.8+148.8\exp(-0.033D_c)$ (relative to that after APEC)) shown in Fig. 6b. Compared with that of
before and after APEC, the $E_{ab}$ of BC-containing particles with ~80-200 nm rBC cores during the day decreased by ~6-15%
and ~10-20%, respectively. Our results provided evidence that emission controls could weaken the light-absorption
capability of ambient BC-containing particles. This weakening would have enhanced the effects of emission control
measures during APEC on BC light absorption.
The reduction in both the rBC mass concentration and the light-absorption capability of ambient BC-containing particles
during APEC revealed a decrease in the light absorption of BC aerosols caused by emission control measures. As shown in
Fig. 7, we calculated the theoretical reduction in the light absorption coefficient of BC during APEC based on the



simultaneous reduction in the mass concentration and light-absorption capability of BC ($\sigma_{ab,with}$), and the reduction in
absorption coefficient calculated without considering the weakened light-absorption capability of BC-containing particles
due to emission reduction ($\sigma_{ab,without}$) was also obtained. The comparison between the reductions in $\sigma_{ab,with}$ and $\sigma_{ab,without}$ could
separate the contributions of a decrease of rBC mass concentration and a weakening of BC light-absorption capability to the
reduction in light absorption during APEC.
Considering the reductions in both the mass concentration and light-absorption capability of BC due to the emission
control measures, the daytime light absorption of BC-containing particles (i.e., $\sigma_{ab,with}$) decreased by ~41% and ~70% during
APEC compared to that before and after APEC, respectively. However, the $\sigma_{ab,without}$ of BC during APEC decreased by ~34%
and ~63% relative to that before and after APEC, respectively (Fig. 7). The difference between the reductions in $\sigma_{ab,with}$ and
$\sigma_{ab,without}$ indicated that the reduction in the rBC concentration contributed ~83% and ~90% of the reduction in BC light
absorption during APEC compared to that before and after APEC, respectively, while the weakening of the BC light-
absorption capability contributed ~17% and ~10%, respectively. On average, the light absorption of BC-containing particles
at daytime during APEC decreased by ~56% compared with before and after APEC, of which ~48% was contributed by the
reduction in the mass concentration of rBC and the remain ~8% was controlled by the weakening of BC light-absorption
capability. These results imply that reductions in the emissions of multiple pollutants (i.e., BC and precursors of secondary
species) in China could benefit air quality and climate due to significantly lowering the light absorption of BC, which was
driven by the reductions in both rBC mass concentration and light-absorption capability of BC-containing particles.
**4 Discussion**
Based on a comparison of the observations before, during and after APEC, we found that the emission control measures
successfully reduced both the rBC mass concentration and the light-absorption capability (i.e., $E_{ab}$) of BC-containing
particles, resulting in a significant decrease in the light absorption of BC. The mechanism underlying the effect of the
emission reductions during APEC on BC light absorption is summarized in Fig. 8. Emission control measures reduce the
amount of both BC and co-emitted secondary aerosol precursors present in the atmosphere. The presence of lower amounts
of secondary particle precursors in the atmosphere weakens the chemical formation of secondary aerosol components,
suppressing the condensation of secondary species on BC surfaces. Less coating material on BC can weaken the lensing
effect, which leads to a weakening of the light-absorption capability for BC-containing particles. Simultaneous reductions in
the mass concentration and light-absorption capability of BC can result in a much lower light absorption of BC during APEC
compared to that before and after APEC.
In China, a series of emission controls measures have been implemented in pollution regions (e.g., Jing-Jin-Ji region),
aiming to increase the number of clean days and decrease the number of haze days. This comparison between periods with
and without emission controls measures may illustrate the differences between clean and polluted periods. In terms of
different pollution levels in China, our findings imply that a clean period is characterized by not only a lower BC mass




concentration but also a weaker light-absorption capability of BC-containing particles compared to that in polluted periods.
In our previous study (Zhang et al., 2018), we found that the light-absorption capability of ambient BC-containing particles
observed in Beijing was enhanced by an increase in pollution levels, resulting in an amplification of BC light absorption
under polluted conditions. The present work clearly demonstrates that emission control measures can reduce this
amplification effect by decreasing the light-absorption capability of BC-containing particles. Moreover, this work can
explain how emission control measures reduce the amplification effect, namely, by slowing the aging of BC through a
reduction in co-emitted secondary aerosol precursors (e.g., $SO_2$, $NO_x$ and VOCs).
The simultaneous reductions in the mass concentration and light-absorption capability of BC due to emission controls
confirmed the suggestions of previous studies that BC emission reductions could achieve multiple benefits, i.e.,
simultaneously controlling air pollution and protecting the climate (Ding et al., 2016). Furthermore, our study implies that
the air quality and climate co-benefits from multi-pollutant emission controls are enhanced by the weakened light-absorption
capability of BC-containing particles. In terms of air quality improvement, the weakened light-absorption capability plays an
important role in both the direct and indirect effects of BC. Weakened light-absorption capability can directly lower the light-
absorbing efficiency of BC aerosols in the atmosphere, resulting in more solar light radiation reaching the surface, and the
weakened light-absorption capability of ambient BC-containing particles can indirectly mitigate air pollution by improving
the planetary boundary layer (PBL) suppression driven by the dome effect of BC (Ding et al., 2016; Wang et al., 2017). On
the other hand, an enhanced reduction in climate warming can be attributed to a smaller direct radiative forcing from BC
aerosols due to a weaker light-absorption capability of atmospheric BC-containing particles. The importance of the
weakened light-absorption capability of BC highlighted in our study provides clues for the management of air quality and
climate change. The emission controls of multiple pollutants including BC and co-emitted secondary aerosol precursors may
be an efficient way to simultaneously mitigate air pollution and climate warming.

## 5 Concluding remarks

The effects of emission reductions on the light absorption of BC-containing particles are not only controlled by the reduction
in the BC mass concentration but also dependent on the change in their light-absorption capability. The decrease in the BC
mass concentration due to emission control measures is well known. However, the impact of emission reduction on the light-
absorption capability of BC-containing particles remains unclear due to a lack of available observations. The 2014 APEC
meeting in Beijing, China, provides an invaluable opportunity to measure the variations in the light-absorption capability of
ambient BC-containing particles due to emission reductions. In this work, based on in situ measurements at an urban site in
Beijing before, during and after APEC using a SP2, we explored whether and how emission control measures in China
influence the light-absorption capability of ambient BC-containing particles. Note that this comparative study focused on the
pollution episodes before, during and after APEC.
We found that the emission control measures successfully lowered the aging degree (i.e., $m_{NR-CM}/m_{rBC}$) of BC-containing



particle. The $m_{NR-CM}/m_{rBC}$ ratio of BC-containing particles with ~80-200 nm rBC cores during APEC decreased by ~10-30%
and ~31-53% compared to that before and after APEC, respectively. The reduction in $m_{NR-CM}/m_{rBC}$ ratio of BC-containing
particles increased with decreasing rBC core size, following an exponential function. The size-dependent reduction in $m_{NR-CM}/m_{rBC}$
$_{CM}/m_{rBC}$ ratio of BC-containing particles indicated that emission reduction was more effective for slowing the aging of
smaller rBC particles. The reduction in $m_{NR-CM}/m_{rBC}$ ratio of BC-containing particles during APEC relative to that before and
after APEC showed a pronounced diurnal cycle, with maxima at ~14:00-17:00 LT and ~10:00-12:00, respectively. The
decreased ageing of BC-containing particles during APEC was mainly driven by a reduction in chemical production (i.e.,
oxidation products such as sulfate and nitrate) on the surface of BC due to less amounts of secondary aerosol precursors
(e.g., the $NO_2$ concentration during APEC decreased by ~34% and ~45% compared with that before and after APEC,
respectively, and the corresponding $SO_2$ concentration decreased by ~35% and ~67% during APEC, respectively) present in
the atmosphere during BC aging. The reduction in $m_{NR-CM}/m_{rBC}$ ratio of BC-containing particles during APEC relative to that
before and after APEC increased with the reduction in the concentrations of $NO_2$ and $SO_2$.
Due to the lower amount of coating materials on BC surfaces during APEC, the light-absorption capability (i.e., $E_{ab}$) of
BC-containing particles with ~80-200 nm rBC cores during the day decreased by ~6-15% and ~10-20% compared to that
before and after APEC, respectively. The weakened light-absorption capability of BC-containing particles enhanced the
reduction in BC light absorption due to the emission control measures. When considering the reduction in both the mass
concentration and light-absorption capability of BC-containing particles during the day during APEC, the theoretical light
absorption (i.e., $\sigma_{ab}$) decreased by ~41% and ~70% compared to that before and after APEC, respectively. However, the
reduced light absorption of BC during the day caused by the decrease in the BC mass concentration during APEC compared
to that before and after APEC was estimated to be ~34% and ~63%, respectively. Therefore, ~10-20% of the reduction in the
daytime light absorption of BC-containing particles during APEC relative to that before and after APEC could be attributed
to the weakened light-absorption capability. Our study revealed that reductions in the emissions of multiple pollutants (i.e.,
BC, $NO_2$ and $SO_2$) could reduce the light-absorption capability of BC. Weakened light-absorption capability of BC due to
emission controls further confirmed the suggestions of previous studies that BC emission reductions can achieve multiple
benefits, i.e., simultaneously controlling air pollution and protecting the climate (Ding et al., 2016; Peng et al., 2016; Zhang
et al., 2018). Our study then implied that the air quality and climate co-benefits from multi-pollutant emission control could
be enhanced by the weakened light-absorption capability of BC-containing particles.
**Acknowledgments**
This work was funded by the National Natural Science Foundation of China (41625020 and 41571130035).





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



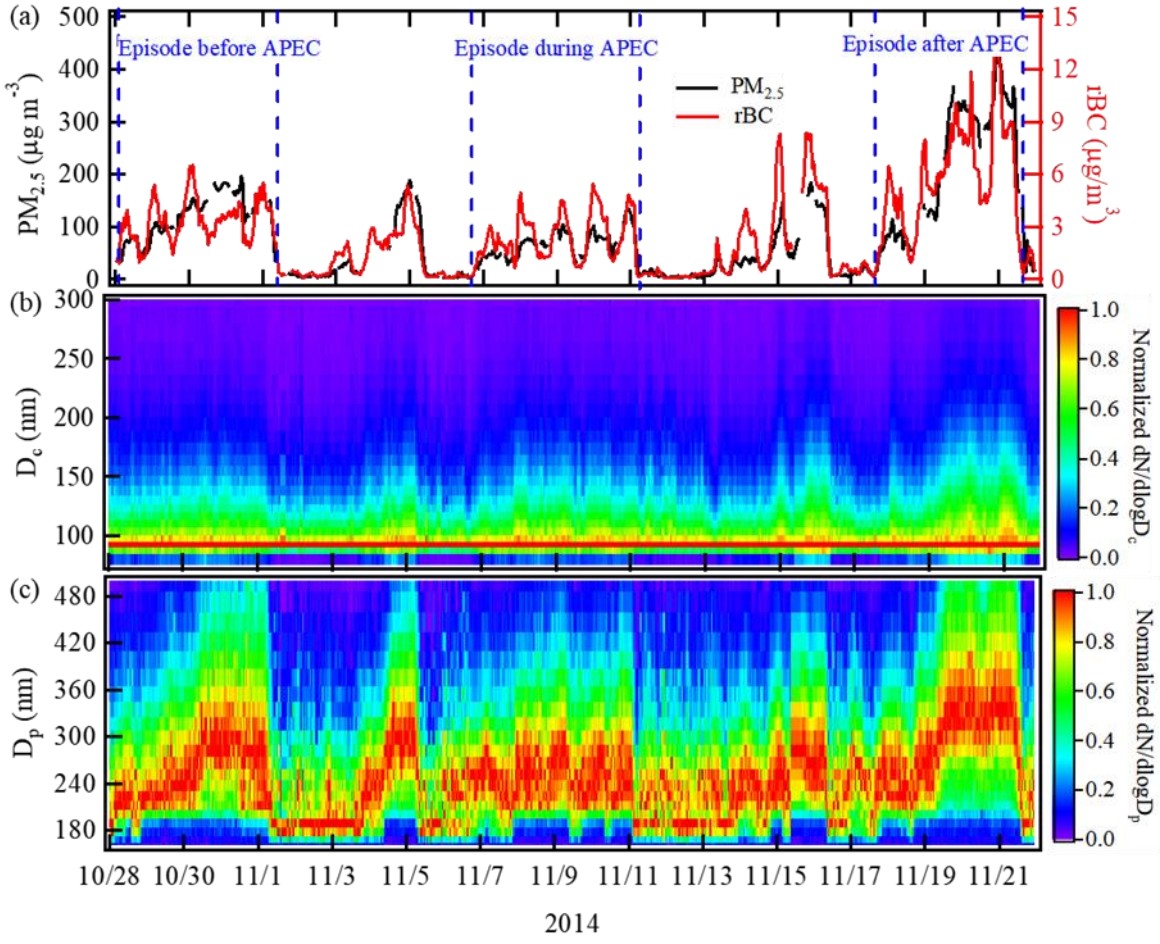

**Figure 1.** Time series of (a) the mass concentrations of $PM_{2.5}$ and rBC and the number size distribution of (b) rBC cores ($D_c$)

and (c) whole BC-containing particles ($D_p$).





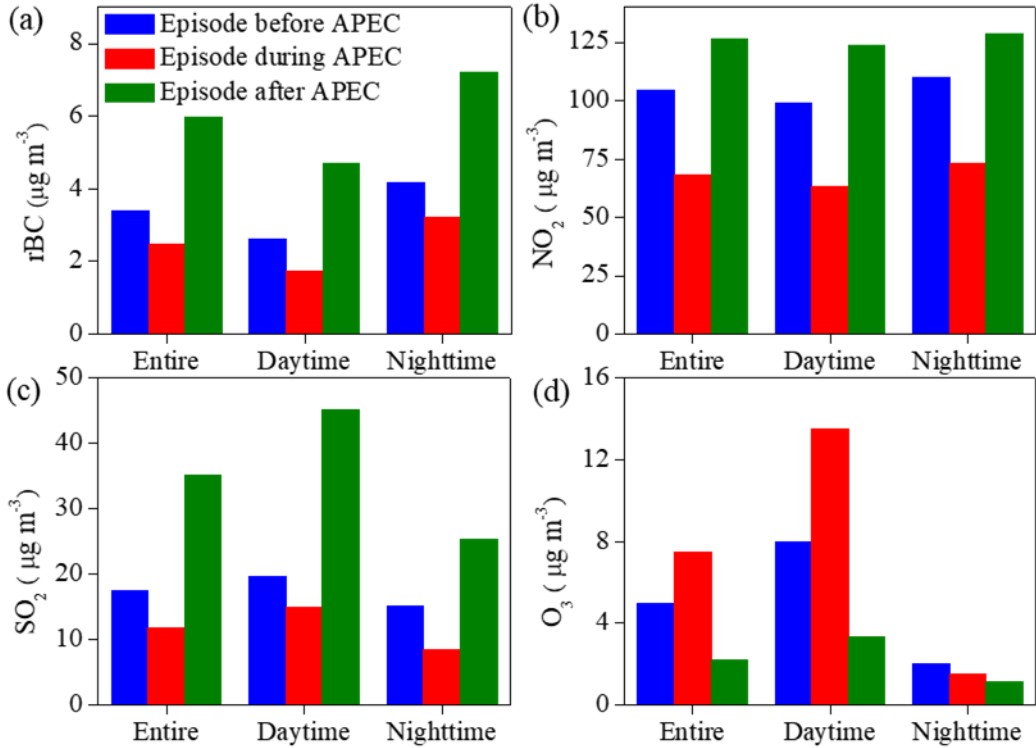

**Figure 2.** The mass concentrations of (a) rBC, (b) NO$_2$, (c) SO$_2$ and (d) O$_3$ for the pollution episodes before, during and after APEC. We

separated the entire data sets into daytime (7:00 LT to 19:00 LT) and nighttime (19:00 LT to 7:00 LT of the following day) sets.

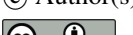


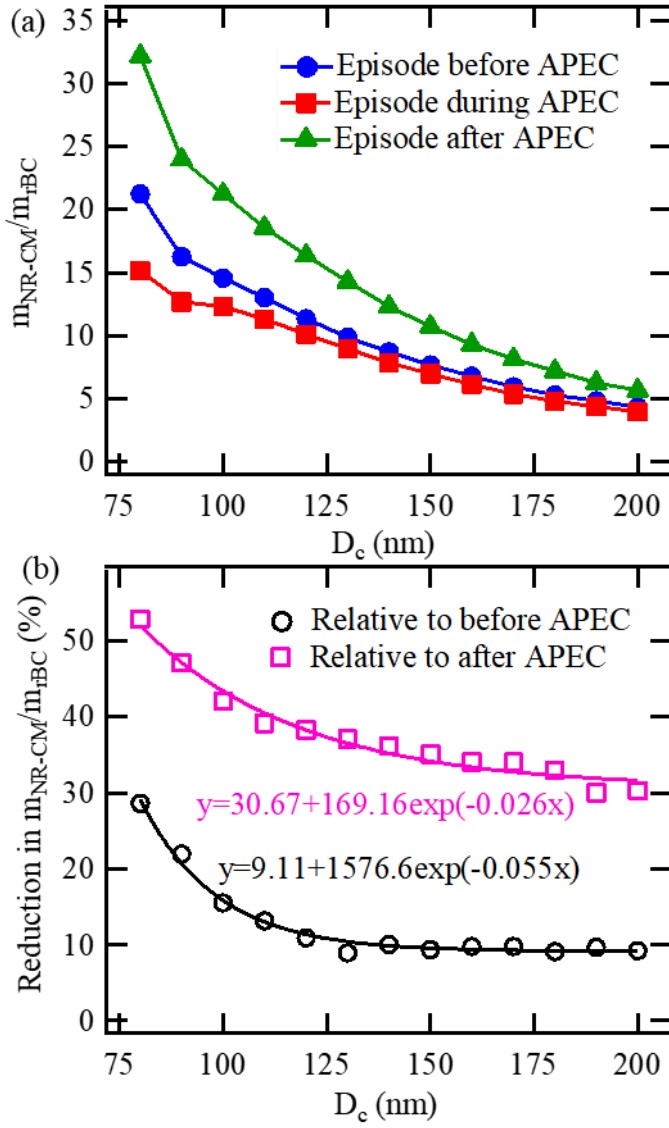

2 **Figure 3.** Comparison of the aging degree of BC-containing particles for the pollution episodes before, during and after APEC: (a)

3 $m_{\text{NR-CM}}/m_{\text{rBC}}$ ratio of BC-containing particles and (b) the reduction in $m_{\text{NR-CM}}/m_{\text{rBC}}$ ratio of BC-containing particles during

4 APEC relative to that before and after APEC.



**Figure 4.** Diurnal cycle of the normalized reduction in $m_{NR-CM}/m_{rBC}$ ratio of BC-containing particles for the pollution episode

during APEC relative to that (a) before and (b) after APEC. (c) Diurnal cycle of $O_3$ concentration for the pollution episodes

before, during and after APEC




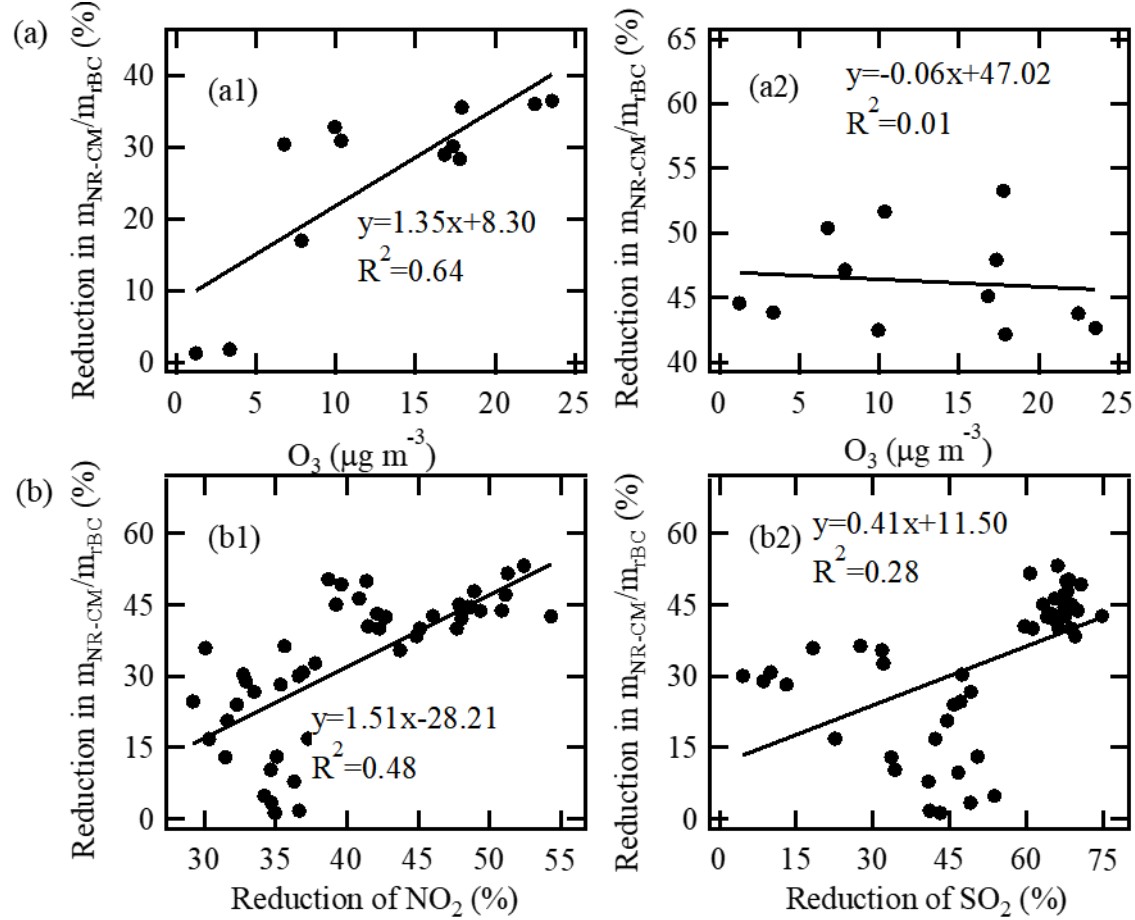

**Figure 5.** (a) Correlation between the reduction in $m_{NR\text{-}CM}/m_{rBC}$ ratio of BC-containing particles for the pollution episode during APEC relative to that (a1) before and (a2) after APEC and the daytime (7:00-19:00) $O_3$ concentration during APEC. (b) Correlation between the reduction in $m_{NR\text{-}CM}/m_{rBC}$ during APEC relative to that before and after APEC and the corresponding reduction in the concentrations of (b1) $NO_2$ and (b2) $SO_2$.



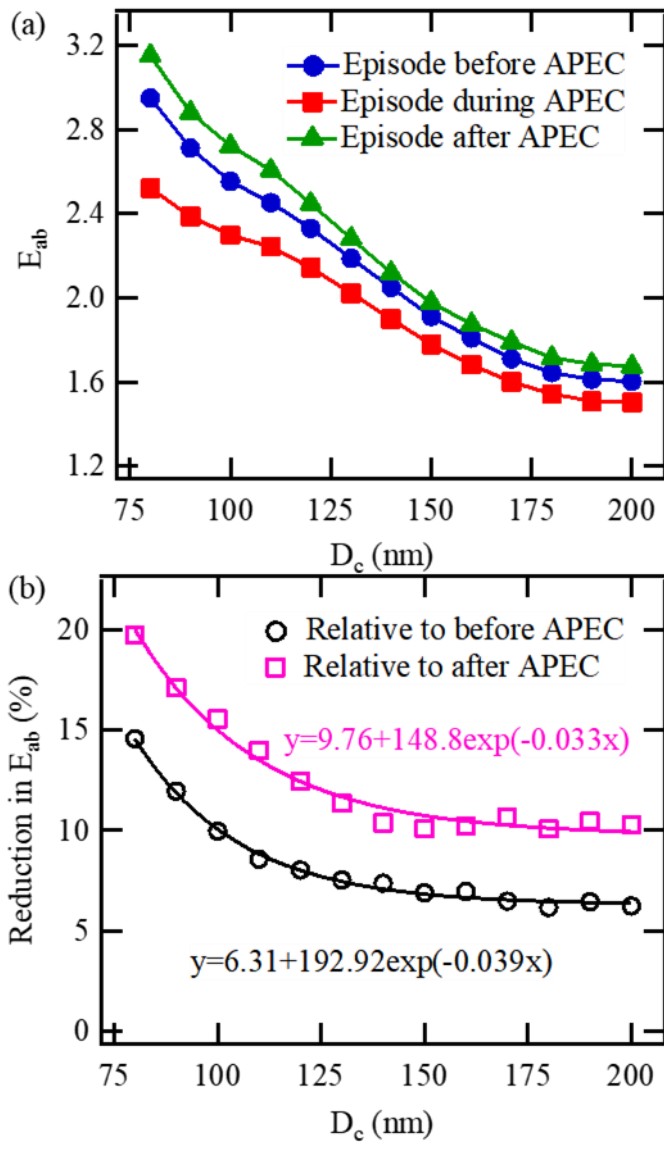

**Figure 6.** Comparison of the light-absorption capability of BC-containing particles during the day for the pollution episodes
before, during and after APEC: (a) light absorption enhancement ($E_{ab}$) of BC-containing particles and (b) the reduction in $E_{ab}$ of
BC-containing particles during APEC relative to that before and after APEC.



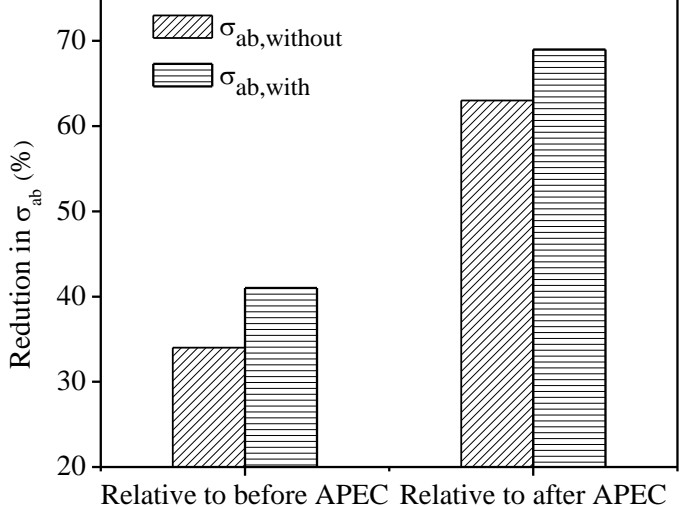

**Figure 7**. Reduction in the absorption coefficients ($\sigma_{ab}$) at 550 nm of BC-containing particles observed in the pollution episode during APEC relative to that before and after APEC. The $\sigma_{ab,with}$ and $\sigma_{ab,without}$ values represent $\sigma_{ab}$ with/without, respectively, considering the differences of light-absorption capability of ambient BC-containing particles among the episodes before, during and after APEC.





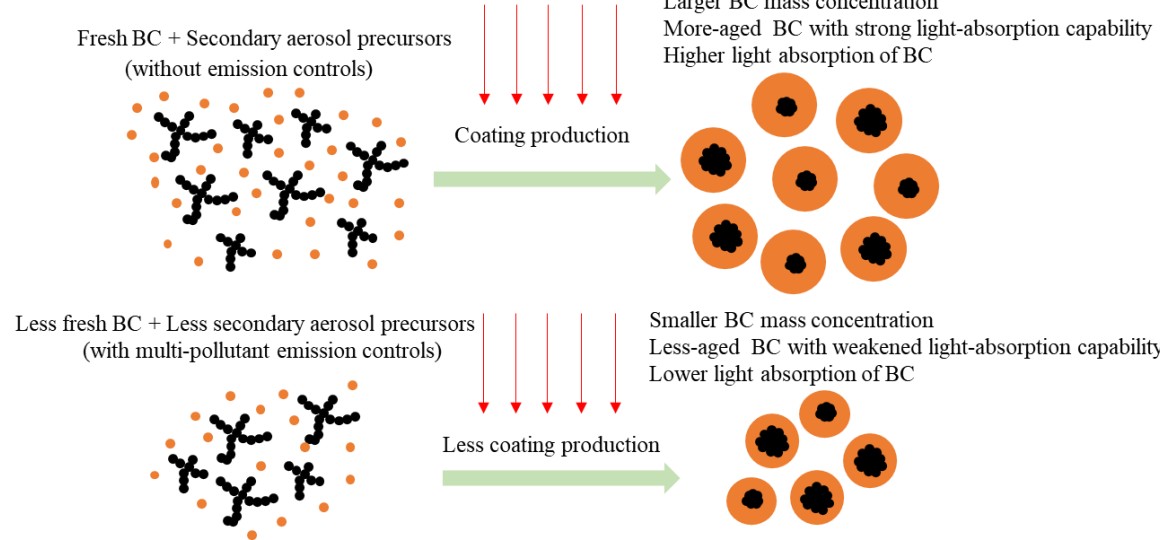

3    **Figure 8.** Conceptual scheme of the reduction in light absorption of BC-containing particles due to multi-pollutant emission

4    controls.