# Peer review of "Reduction in black carbon light absorption due to multi-pollutant"

_Atmospheric Chemistry and Physics, 2018_

## Referee Comment (RC1) · Anonymous Referee #1 · 17 Apr 2018

The authors measured rBC particles with a single particle soot photometer during APEC, and discussed the effects of multi-pollutant emission reductions on the BC light absorption. The results are interesting, and the science of this work sounds good. However there some potential issues with the data analysis and conclusions. Moreover, the English needs to be further improved. I believe this manuscript can be considered for publication in ACP after minor revision.

Page 2 Lines 17–18: Previous studies have demonstrated that a majority of the freshly emitted BC particles are internally mixed from biomass burning emissions (e.g., www.sciencedirect.com/science/article/pii/S135223101830133X). Thus, this ex-

pression should be revised.

Page 3 Lines 26-27: This sentence is hard to understand. How long is the BC lifetime that the authors considered in this study?

Page 4 Lines 25-27: The ratio of mNR-CM/mrBC was calculated based on several assumptions, e.g, RI, density, and core-shell structure, thus, it is better to add some discussion about uncertainties of this method.

Page 5 Lines 10–14: According to Zhang et al. (2018), the authors also used AE33 measurements during this campaign. It is better to compare the calculated light absorption coefficients (based on Eq. 3) with the AE33 measured values.

Page 5 Lines 17–18: What's the standard used to define the pollution episodes?

Page 6 Lines 5-6: The emission control can reduce the concentrations of NO2 and SO2. That's right, and a lot of APEC publications have demonstrated. However, the authors can not obtain the statement of BC coating materials being affected by emission controls from this, though the authors prove this may be right in the following discussion (Fig. 5).

Page 6 Lines 7-9: It's better to add previous studies in Beijing (e.g, http://iopscience.iop.org/article/10.1088/1748-9326/aa64ea/meta) to explain the importance of photochemical reactions in BC aging process.

Page 6 Lines 7-19: The large reductions in the daytime levels of rBC and coating precursors may be due to the higher boundary layer, which favors diffusion of pollutants. How to evaluate the impacts of meteorological conditions and emission reductions on the rBC coatings?

Page 6 Line 23-24: The similar rBC core size during different periods may be due to the similar emission sources, but the statement of "similar atmospheric processes" is not right. The atmospheric processes not only affect rBC core but also the BC-containing particles. Thus, this statement should be reworked.

Page 6 Lines 28-32: Based on the AMS measurements in Beijing, primary organic aerosol (POA) is also an important species. Therefore, POA may be also an important contributor to BC coatings.

Page 8 Lines 20–21: Here it would be good to know if the reductions are statistically significant.
* * *

---

## Referee Comment (RC2) · Anonymous Referee #2 · 25 Apr 2018

General comments:

The present manuscript reports variations of the refractory BC (rBC) particles measured by a single particle soot photometer before/during/after APEC summit. The shell diameter of rBC-containing particles were determined according to Mie scattering theory presuming that all rBC particles were in shell-core configuration. The objective of this study is to evaluate the effect of emission control measures on the mixing state of rBC as well as their light absorbing properties, and concluded that coating matters on the rBC core decrease as a result of emission control of pollution precursors such as $SO_2$ and $NO_2$. In general, the paper is clear logically and well written; however

deficiencies of this study is the absorption enhancement (Eab) is estimated based on calculation, not measurement. Direct measurement of optical properties of rBC particles is essential to better understand the radiative effect of rBC containing particles in Beijing. The paper could be considered for publication after several issues are carefully clarified, as follows:

Page 4 line 16: The authors assumed that all the rBC containing particles were in spherical shape, please make sure they are true or not. As known, on-road vehicle emission is one of the important sources for rBC particles in Beijing mega city, freshly emitted rBC may present in non-spherical shape, and they may turn to be spherical with ageing process. The authors are suggested to check the number size distribution and delay time (delta_t) to clarify this.

Page 5 section 2.3.2: As mentioned, Eab is determined on Mie theory, not measurements. Please point out the uncertainty of such calculation related to RI values. Adding an uncertainty in Figure 6 is encouraged.

Page 6 line 11: The authors attribute the decrease in ambient rBC and NO2 on the emission controls. It is encouraged to give more discussion show diurnal variations of their concentration (Sun, Y. et al. "APEC Blue": Secondary Aerosol Reductions from Emission Controls in Beijing. Sci. Rep. 6, 20668; doi: 10.1038/srep20668 (2016), as well as back trajectories of air mass during APEC and non-APEC period to support it.

Page 6 line 24: It is hard to understand the meaning of the sentence "The similar mode of . . . and similar atmospheric processes (coagulation and wet removal) for rBC particles . . . APEC." The predominance of rBC in ∼100 nm range is mostly due to incomplete combustion processes (vehicle engine etc.). It is better to remove it.

Page 7 line 4: "condensational growth was more effect". Please provide more information to support such statement.

In the third paragraph, Where is "Fig. 4b"? What is the reason for the difference in

diurnal variability of Dc before and after APEC period?

---

## Short Comment (SC1) · 15 May 2018

The authors combined SP2 measurements and Mie theory calculations to provide evidence for the reduction of black carbon (BC) light absorption due to the APEC emission control. The paper is well written. I have one minor suggestion on the uncertainty associated with the calculation/analysis of BC light absorption.

Recent observations (e.g., China et al., 2015; Wang et al., 2017) have shown various complicated BC coating structures/morphology, which are not core-shell. Further modeling studies (e.g., Scarnato et al., 2013; He et al., 2015, 2016) have indicated a large variation in BC absorption and scattering due to the observed complex particle coat-

ing structures/morphology. Thus, assuming a core-shell structure in the present study may lead to uncertainty in the estimate of BC light absorption. It would be helpful if the authors could include these recent studies and add some discussions on this issue.

References

China, S., et al.: Morphology and mixing state of aged soot particles at a remote marine free troposphere site: Implications for optical properties, Geophys. Res. Lett., 42, 1243–1250, doi:10.1002/2014gl062404, 2015.

He, C., et al.: Variation of the radiative properties during black carbon aging: theoretical and experimental intercomparison, Atmos. Chem. Phys., 15, 11967-11980, doi:10.5194/acp-15-11967-2015, 2015.

He, C., et al.: Intercomparison of the GOS approach, superposition T-matrix method, and laboratory measurements for black carbon optical properties during aging, J. Quant. Spectrosc. Radiat. Transf., 184, 287–296, doi:10.1016/j.jqsrt.2016.08.004, 2016.

Scarnato, B. V., et al.: Effects of internal mixing and aggregate morphology on optical properties of black carbon using a discrete dipole approximation model, Atmos. Chem. Phys., 13, 5089–5101, doi:10.5194/acp-13-5089-2013, 2013.

Wang, Y., et al.: Fractal dimensions and mixing structures of soot particles during atmospheric processing, Environ. Sci. Technol. Lett., 4, 487-493, doi:10.1021/acs.estlett.7b00418, 2017.

---

## Author Comment (AC1) · 27 Jun 2018

Anonymous Referee #1:

The authors measured rBC particles with a single particle soot photometer during APEC, and discussed the effects of multi-pollutant emission reductions on the BC light absorption. The results are interesting, and the science of this work sounds good. However there some potential issues with the data analysis and conclusions. Moreover, the English needs to be further improved. I believe this manuscript can be considered for publication in ACP after minor revision.

We would like to thank the reviewer for the valuable and constructive comments, which helps us to improve the manuscript. Listed below are our responses to the comments point-by-point, as well as the corresponding changes made to the revised manuscript. The reviewer's comments are marked in black and our answers are marked in blue, and the revision in the manuscript is further formatted as '*Italics*'.

1. Page 2 Lines 17–18: Previous studies have demonstrated that a majority of the freshly emitted BC particles are internally mixed from biomass burning emissions (e.g., www.sciencedirect.com/science/article/pii/S135223101830133X). Thus, this expression should be revised.

   **Response:** Thanks. We have revised the sentence as "*When fresh BC particles are emitted from incomplete combustion (e.g., traffic emission) other than biomass burning (Wang et al., 2018; Pan et al., 2017), they are most likely externally mixed with other aerosol components.*"

2. Page 3 Lines 26-27: This sentence is hard to understand. How long is the BC lifetime that the authors considered in this study?

   **Response:** Thanks for the comment. In this study, we defined the transition periods (November 3-5 and 13-16, 2014) considering the lifetime (~3 d) of BC during the campaign period. Figure R1 in the response (Fig. S2 in the revised manuscript) shows a similar distribution of effective emission intensity (EEI,

defined by Lu et al. (2012)) of BC over the site during 3, 5 and 7 days, revealing that the BC transported to the site was mainly from emission within 3 days. The EEI analysis indicated that the BC particles over the site during the campaign period have a lift time of ~3 d. For "APEC blue", the emission control measures were implement on November 3-12, 2014. Considering the ~3 d lifetime, the BC transported to the site during the pollution episode on November 3-5 were the mixtures of particles that were emitted before and during APEC. Similarly, the BC transported to the site during the pollution episode on November 13-16 were the mixtures of particles that were emitted during and after APEC. To clearly distinguish BC characteristics with and without emission control measures, we exclude this two periods (November 3-5 and 13-16, 2014) in this study.

[Figure]

Figure R1 (Fig. S2 in the revised manuscript). Spatial distribution (0.25°×0.25°) of the effective emission intensity (EEI, defined by Lu et al. (2012)) for BC transported to the observation site (40º00'17" N, 116º19'34" E) based on 3 days, 5 days and 7 days back-trajectory. The EEI calculation was stated in our previous study (Zhang et al. 2018).

To make this point clear, we have revised the sentence as "*and two transition periods (November 3-5 and 13-16, 2014), which were not discussed in this work considering that we could not distinguish the BC particles transported to the site during these days characterized by enforcement of emission control measures or not (Fig. S2 and the associated discussion in the supplementary information).*"

3. Page 4 Lines 25-27: The ratio of mNR-CM/mrBC was calculated based on several assumptions, e.g, RI, density, and core-shell structure, thus, it is better to add some discussion about uncertainties of this method.

**Response:** Thanks to the reviewer for raising this concern. Following the reviewer's suggestion, we have add some discussion about uncertainties of Mie calculation. In our previous work (Zhang et al., 2018), we have estimated the uncertainties from the assumptions in Mie calculation. These assumptions were also used in this work to calculate the $m_{NR-CM}/m_{rBC}$. Following the reviewer's suggestion, we have added the related discussion in the revised manuscript, as *"The uncertainty of Mie calculation from the assumptions (e.g., RI, density and core-shell structure) was estimated to be ~10% in our previous work (Zhang et al., 2018)."*

Correspondingly, we have added the uncertainty in the new Fig. 3 (Fig. R2 in the response). The related statements was also added in the caption.

[Figure]

Figure R2 (new Fig. 3b in the revised manuscript). The reduction in $m_{NR-CM}/m_{rBC}$ ratio of BC-containing particles during APEC relative to that before and after APEC. The error bar shown in (b) represents the uncertainties (~10%) from Mie calculation.

4. Page 5 Lines 10–14: According to Zhang et al. (2018), the authors also used AE33 measurements during this campaign. It is better to compare the calculated light absorption coefficients (based on Eq. 3) with the AE33 measured values.

**Response:** Thanks for the comment. Following the reviewer's suggestion, we have compared the calculated light absorption coefficients with the measured values (Fig. R3 in the response). However, we used the MAAP measurements instead of AE33 measurements, because the AE33 measurement was not conducted before APEC.

Correspondingly, the related discussion has been added in the revised manuscript, as "*Figure 7a shows the measured and theoretical light absorption coefficient ($\sigma_{ab}$) of BC-containing particles during the campaign period. The measured $\sigma_{ab}$ revealed that the daytime light absorption of BC-containing particles in the pollution episode during APEC decreased by ~42% and ~68% compared with those in pollution episode before and after APEC, respectively. This decrease could attributed to reduction in both the rBC mass concentration and the light-absorption capability of ambient BC-containing particles. In order to separate the contributions of a decrease of rBC mass concentration and a weakening of BC light-absorption capability to the reduction in light absorption during APEC, we calculated the theoretical reduction in $\sigma_{ab}$ of BC-containing during APEC with and without considering the weakened light-absorption capability of BC-containing particles due to emission reduction ($\sigma_{ab,with}$ and $\sigma_{ab,without}$ respectively). When considering the simultaneous reduction in the mass concentration and light-absorption capability of BC, the calculated reduction in daytime $\sigma_{ab}$ of BC-containing during APEC related to non-APEC period showed a good agreement with ones obtained from MAAP measurements (Fig. 7b). This agreement demonstrated that the decrease in the light absorption of BC-containing particles depended not only on the reduction of BC mass concentration, but also on the weakening of their light-absorption capability.*"

[Figure]

Figure R3 (new Fig. 7 in the revised manuscript). (a) The light absorption coefficient ($\sigma_{ab}$) at 670 nm. (b) Reduction in the absorption coefficients ($\sigma_{ab}$) of BC-containing particles observed in the pollution episode during APEC relative to that before and after APEC. The correlation between the calculated $\sigma_{ab}$ ($\sigma_{ab, calculated}$) using Mie theory combined with SP2 measurements and the measured $\sigma_{ab}$ ($\sigma_{ab, measured}$) by the MAAP is also shown in (a). The $\sigma_{ab,with}$ and $\sigma_{ab,without}$ values represent $\sigma_{ab, calculated}$ with/without, respectively, considering the differences of light-absorption capability of ambient BC-containing particles among the episodes before, during and after APEC.

5. Page 5 Lines 17–18: What's the standard used to define the pollution episodes?

**Response:** We thank the reviewer for raising this question. Following Sun et al. (2016), the pollution episodes were briefly separated by clean days through the study period (Fig. 1 in the manuscript). Air flows were predominately southerly and meteorological conditions were generally stagnant (low wind speed and high relative humidity) during these episodes. Because emission controls were mainly implemented in cities to the south and east of Beijing, we use these episodes to study the impact of regional emission controls on aerosol chemistry in megacity Beijing. This approach isolates the influences of clean periods with air masses from the north and northwest where far fewer emission controls were implemented.

    To make it clear, the related statement has been added in the revised manuscript, as "*Three pollution episodes (briefly separated by clean days) on October 28-November 1, November 6-11 and November 17-21 were observed before, during and after APEC, respectively. Following APEC study in Sun et al. (2016), we focused on comparing the BC characteristics among the pollution episodes to investigate the effect of emission reduction. During the three pollution episodes, the air masses over the site were mainly from the south and east of Beijing (Fig. S3) where emission control measures were implemented during APEC. On the other hand, the pollution episodes in Beijing were characterized by low wind speed and planetary boundary layer, as well as high relative humidity (Sun et al. 2016; Zheng et al. 2015).*"

6. Page 6 Lines 5-6: The emission control can reduce the concentrations of NO2 and SO2. That's right, and a lot of APEC publications have demonstrated. However, the authors can not obtain the statement of BC coating materials being affected by emission controls from this, though the authors prove this may be right in the following discussion (Fig. 5).

**Response:** Thanks for the comments. Here, we stated that reduction in the concentrations of $NO_2$ and $SO_2$ might affect the production of BC coating

materials. Whether the reduction of precursor of secondary aerosols (e.g., $NO_2$ and $SO_2$) will affect the coating materials on the BC is complex, which not only depends on the decrease in BC amount versus secondary aerosols but also controlled by secondary components condensed on BC-containing versus non-BC containing particles. As expected, in the following section (3.2), we found the reduction in BC coating materials during APEC compared with that before/after APEC based on SP2 measurement.

To make it clear, we have revised the sentence as "*Previous studies have identified a reduction in the concentrations of secondary components (e.g., sulfate and nitrate) in aerosols during APEC compared to that before and after APEC (Zhang et al., 2016a; Han et al., 2015). However, the change of coating materials on the BC due to the reduction of secondary components was complex, which not only determined by the decrease in BC versus secondary components, but also depend on secondary components condensed on BC-containing versus non-BC particles.*"

7. Page 6 Lines 7-9: It's better to add previous studies in Beijing (e.g, http://iopscience.iop.org/article/10.1088/1748-9326/aa64ea/meta) to explain the importance of photochemical reactions in BC aging process.

    **Response:** Thanks for the suggestion. Following the reviewer's suggestion, we have added previous studies in Beijing (Wang et al., 2017; Metcalf et al., 2013; Zhang et al., 2014; Peng et al., 2016) to explain the importance of photochemical reactions in BC aging process. The related discussion was as "*Previous studies have pointed out the importance of photochemical reactions in BC aging process (Wang et al., 2017; Metcalf et al., 2013; Zhang et al., 2014; Peng et al., 2016), indicating that changing the daytime concentrations of rBC and coating precursors might play a more important role in affecting BC aging than altering the nighttime concentrations.*"

8. Page 6 Lines 7-19: The large reductions in the daytime levels of rBC and coating

precursors may be due to the higher boundary layer, which favors diffusion of pollutants. How to evaluate the impacts of meteorological conditions and emission reductions on the rBC coatings?

**Response:** We thank the reviewer for raising this question. In order to segregate possible meteorological effects and to quantify approximately the influence of the emission reductions on BC coatings, we picked up observational data during pollution episodes before, during and after APEC, in which the local meteorological conditions were similar. On the other hand, we have given diurnal variations of the rBC, $NO_2$ and $SO_2$ concentration and the PBL for the pollution episodes before, during and after APEC (Fig. R4 in the response and Fig. S4 in the supplement), as well as back trajectories of air mass during APEC and non-APEC period (Fig. R5 in the response and Fig. S3 in the supplement) to support the decrease in ambient rBC, $NO_2$ due to the emission controls.

The related discussion was added "*Figure S4 shows that the diurnal variations of the rBC, $NO_2$ and $SO_2$ concentration and the PBL during the pollution episodes before, during and after APEC. Comparing the diurnal variations between the rBC concentration and the PBL revealed that the rBC concentrations during the pollution episodes were dominated by the PBL. However, the precursor concentration of secondary aerosol (i.e., $NO_2$ and $SO_2$) during the pollution episodes exhibited different diurnal variations with a peak at noontime and early afternoon, which was most likely attributed to regional transport. The back-trajectory analysis (Fig. S3) revealed that the air mass during the pollution episodes was mainly from polluted regions (i.e., Hebei and Tianjin). This indicated that regional emission controls would reduce the pollutant (i.e., rBC, $NO_2$ and $SO_2$) concentration in Beijing under polluted conditions. The Sun et al., (2016) has demonstrated significant reductions in the precursors of secondary aerosol during APEC compared to those in non-APEC period due to emission controls during over a regional scale (i.e., Beijing and adjacent areas). The similar PBL (Fig S4) during the pollution episodes before, during and after APEC further identified the important contribution of emission reduction to the*

*decrease of rBC, NO₂ and SO₂ concentration during APEC."*

[Figure]

Figure R4 (Fig. S4 in the revised manuscript). Diurnal variations of the rBC, NO₂ and SO₂ concentrations and the PBL for the pollution episodes before, during and

after APEC.

[Figure]

Figure R5 (Fig. S3 in the revised manuscript). Spatial distribution (0.25°×0.25°) of the effective emission intensity (EEI, defined by Lu et al. (2012)) for BC transported to the observation site (40º00'17" N, 116º19'34" E) for the pollution episodes before, during and after APEC. The EEI calculation was stated in our previous study (Zhang et al. 2018).

9. Page 6 Line 23-24: The similar rBC core size during different periods may be due to the similar emission sources, but the statement of "similar atmospheric processes" is not right. The atmospheric processes not only affect rBC core but also the BC-containing particles. Thus, this statement should be reworked.

   **Response:** Thanks to the reviewer to point this out. Following the reviewer's suggestion, we have deleted the statement of "similar atmospheric processes" and the sentence was revised as "*The similar modes of the rBC cores could have resulted from similar emission sources for BC-containing particles observed before, during and after APEC.*"

10. Page 6 Lines 28-32: Based on the AMS measurements in Beijing, primary organic aerosol (POA) is also an important species. Therefore, POA may be also an important contributor to BC coatings.

    **Response:** Thanks for the comment. We have revised the sentence in the revised manuscript, as "*In the pollution episodes before and after APEC, the particle size of the whole BC-containing particles exhibited sustained growth from ~180 nm to ~320 and ~400 nm, respectively, which could be attributed to the gradual*

*coagulation and condensation of other species (i.e., primary aerosol and secondary components) on the BC surface.*"

11. Page 8 Lines 20–21: Here it would be good to know if the reductions are statistically significant.

**Response:** We thank the reviewer for raising this question. Following the reviewer's suggestion, we have shown the statistical results in the new Fig. 6 (Fig. R6 in the response).

[Figure]

Figure R6 (new Fig. 6 in the revised manuscript). Comparison of the light-absorption capability of BC-containing particles during the day for the

pollution episodes before, during and after APEC: (a) light absorption enhancement ($E_{ab}$) of BC-containing particles and (b) the reduction in $E_{ab}$ of BC-containing particles during APEC relative to that before and after APEC. The error bar shown in (b) represents the uncertainties (~15%) of $E_{ab}$ from Mie calculation.

**References:**

Han, T., Xu, W., Chen, C., Liu, X., Wang, Q., Li, J., Zhao, X., Du, W., Wang, Z., and Sun, Y.: Chemical apportionment of aerosol optical properties during the Asia-Pacific Economic Cooperation summit in Beijing, China, *J. Geophys. Res.-Atmos.*, 120, 12, 281-212, 295, dio:10.1002/2015JD023918, 2015.

Lu, Z., Streets, D. G., Zhang, Q., and Wang, S.: A novel back-trajectory analysis of the origin of black carbon transported to the Himalayas and Tibetan Plateau during 1996–2010, *Geophys. Res. Lett.*, 39, 2012.

Metcalf, A. R., Loza, C. L., Coggon, M. M., Craven, J. S., Jonsson, H. H., Flagan, R. C., and Seinfeld, J. H.: Secondary Organic Aerosol Coating Formation and Evaporation: Chamber Studies Using Black Carbon Seed Aerosol and the Single-Particle Soot Photometer, *Aerosol Sci. Technol.*, 47, 326-347, 2013.

Pan, X., Kanaya, Y., Taketani, F., Miyakawa, T., Inomata, S., Komazaki, Y., Tanimoto, H., Wang, Z., Uno, I., and Wang, Z.: Emission characteristics of refractory black carbon aerosols from fresh biomass burning: a perspective from laboratory experiments, *Atmos. Chem. Phys.*, 17, 13001-13016, 2017.

Peng, J., Hu, M., Guo, S., Du, Z., Zheng, J., Shang, D., Levy Zamora, M., Zeng, L., Shao, M., Wu, Y.-S., Zheng, J., Wang, Y., Glen, C. R., Collins, D. R., Molina, M. J., and Zhang, R.: Markedly enhanced absorption and direct radiative forcing of black carbon under polluted urban environments, *Proc. Natl. Acad. Sci. USA*, 113, 4266-4271, 2016.

Sun, Y., Wang, Z., Wild, O., Xu, W., Chen, C., Fu, P., Du, W., Zhou, L., Zhang, Q., Han, T., Wang, Q., Pan, X., Zheng, H., Li, J., Guo, X., Liu, J., and Worsnop, D. R.: "APEC Blue": Secondary Aerosol Reductions from Emission Controls in Beijing, *Sci.*

*Rep.*, 6, 20668, 2016.

Wang, Q., Cao, J., Han, Y., Tian, J., Zhang, Y., Pongpiachan, S., Zhang, Y., Li, L., Niu, X., Shen, Z., Zhao, Z., Tipmanee, D., Bunsomboonsakul, S., Chen, Y., and Sun, J.: Enhanced light absorption due to the mixing state of black carbon in fresh biomass burning emissions, *Atmos. Environ.*,180, 184-191, 2018.

Wang, Q., Huang R., Zhao, Z., Cao, J., Ni, H., Tie, X., Zhu, C., Shen, Z., Wang, M., Dai, W., Han, Y., Zhang, N., and Prévôt, A.: Effects of photochemical oxidation on the mixing state and light absorption of black carbon in the urban atmosphere of China, *Environ. Res. Lett.*, 12, 044012, 2017.

Zhang, G., Bi, X., He, J., Chen, D., Chan, L. Y., Xie, G., Wang, X., Sheng, G., Fu, J., and Zhou, Z.: Variation of secondary coatings associated with elemental carbon by single particle analysis, *Atmos. Environ.*, 92, 162-170, 2014.

Zhang, J. K., Wang, L. L., Wang, Y. H., and Wang, Y. S.: Submicron aerosols during the Beijing Asia–Pacific Economic Cooperation conference in 2014, *Atmos. Environ.*, 124, Part B, 224-231, 2016a.

Zhang, Y., Zhang, Q., Cheng, Y., Su, H., Li, H., Li, M., Zhang, X., Ding, A., and He, K.: Amplification of light absorption of black carbon associated with air pollution, *Atmos. Chem. Phys. Discuss.*, https://doi.org/10.5194/acp-2017-983, in review, 2018.

Zheng, G. J., Duan, F. K., Su, H., Ma, Y. L., Cheng, Y., Zheng, B., Zhang, Q., Huang, T., Kimoto, T., Chang, D., Pöschl, U., Cheng, Y. F., and He, K. B.: Exploring the severe winter haze in Beijing: the impact of synoptic weather, regional transport and heterogeneous reactions, *Atmos. Chem. Phys.*, 15, 2969-2983, 2015.

---

## Author Comment (AC2) · 27 Jun 2018

Anonymous Referee #2:

We would like to thank the reviewer for the valuable and constructive comments, which helps us to improve the manuscript. Listed below are our responses to the comments point-by-point, as well as the corresponding changes made to the revised manuscript. The reviewer's comments are marked in black and our answers are marked in blue, and the revision in the manuscript is further formatted as '*Italics*'.

**1. General comments**

The present manuscript reports variations of the refractory BC (rBC) particles measured by a single particle soot photometer before/during/after APEC summit. The shell diameter of rBC-containing particles were determined according to Mie scattering theory presuming that all rBC particles were in shell-core configuration. The objective of this study is to evaluate the effect of emission control measures on the mixing state of rBC as well as their light absorbing properties, and concluded that coating matters on the rBC core decrease as a result of emission control of pollution precursors such as $SO_2$ and $NO_2$. In general, the paper is clear logically and well written; however deficiencies of this study is the absorption enhancement ($E_{ab}$) is estimated based on calculation, not measurement. Direct measurement of optical properties of rBC particles is essential to better understand the radiative effect of rBC containing particles in Beijing. The paper could be considered for publication after several issues are carefully clarified, as follows.

**Response:** We thank the reviewer for raising the important issue. Following the reviewer's suggestion, we have added the measured light absorption coefficients to compare with the calculated values (Fig. R1 in the response and new Fig. 7 in the revised manuscript).

[Figure]

Figure R1 (new Fig. 7 in the revised manuscript). (a) The light absorption coefficient ($\sigma_{ab}$) at 670 nm. (b) Reduction in the daytime $\sigma_{ab}$ of BC-containing particles observed in the pollution episode during APEC relative to that before and after APEC. The correlation between the calculated $\sigma_{ab}$ ($\sigma_{ab,\ calculated}$) using Mie theory combined with SP2 measurements and the measured $\sigma_{ab}$ ($\sigma_{ab,\ measured}$) by the MAAP is also shown in (a). The $\sigma_{ab,with}$ and $\sigma_{ab,without}$ values represent $\sigma_{ab,\ calculated}$ with/without, respectively, considering the differences of light-absorption capability of ambient BC-containing particles among the episodes before, during and after APEC.

Correspondingly, the related discussion has been added in the revised manuscript, as "*Figure 7a shows the measured and theoretical light absorption coefficient ($\sigma_{ab}$) of BC-containing particles during the campaign period. The measured $\sigma_{ab}$ revealed that the daytime light absorption of BC-containing particles in the pollution episode during APEC decreased by ~42% and ~68% compared with those in pollution episode*

*before and after APEC, respectively. This decrease could attributed to reduction in both the rBC mass concentration and the light-absorption capability of ambient BC-containing particles. In order to separate the contributions of a decrease of rBC mass concentration and a weakening of BC light-absorption capability to the reduction in light absorption during APEC, we calculated the theoretical reduction in $\sigma_{ab}$ of BC-containing during APEC with and without considering the weakened light-absorption capability of BC-containing particles due to emission reduction ($\sigma_{ab,with}$ and $\sigma_{ab,without}$ respectively). When considering the simultaneous reduction in the mass concentration and light-absorption capability of BC, the calculated reduction in daytime $\sigma_{ab}$ of BC-containing during APEC related to non-APEC period showed a good agreement with ones obtained from MAAP measurements (Fig. 7b). This agreement demonstrated that the decrease in the light absorption of BC-containing particles depended not only on the reduction of BC mass concentration, but also on the weakening of their light-absorption capability.*"

**2. Specific comments**

(1) Page 4 line 16: The authors assumed that all the rBC containing particles were in spherical shape, please make sure they are true or not. As known, on-road vehicle emission is one of the important sources for rBC particles in Beijing mega city, freshly emitted rBC may present in non-spherical shape, and they may turn to be spherical with ageing process. The authors are suggested to check the number size distribution and delay time (delta_t) to clarify this.

**Response:** Thanks to the reviewer to point this out. Following the reviewer's suggestion, we have checked the number distribution of rBC cores ($D_c$) and whole BC-containing particles ($D_p$) (Fig. R2 in the response) to clarify the reasonability of spherical assumption used in Mie calculation in this study. Noted that we focused on investigating the BC-containing particles during pollution episodes. Figure R2 shows that the number distribution of $D_p$ for BC-containing particles

exhibited a peak at 200-400 nm during the pollution episodes, significantly larger than the peak value ($D_c$ of ~95 nm) for number size distribution of bare rBC cores. This revealed fully aged BC-containing particles under polluted conditions. In our previous study (Zhang et al., 2016), we found that the thickly coated BC particles in the north china plain (including Beijing) exhibited near-spherical shape. Therefore, the spherical assumption used in the Mie calculation in this study was reasonable.

[Figure]

Figure R2 (Fig. 1 in the revised manuscript). Time series of the number size distribution of rBC cores ($D_c$) and whole BC-containing particles ($D_p$).

To make this point clear, we have added the Fig. S1 in the supplement and the related statement was "*In this study, we focused on investigating the BC-containing particles during pollution episodes. Under polluted conditions, we have found fully aged BC-containing particles in Beijing, China (Zhang et al., 2018). In our previous study (Zhang et al., 2016), we found that the thickly coated BC particles in the north china plain (including Beijing) exhibited near-spherical shape and a core-shell structure used in the Mie calculation was reasonable.*"

(2) Page 5 section 2.3.2: As mentioned, Eab is determined on Mie theory, not measurements. Please point out the uncertainty of such calculation related to RI values. Adding an uncertainty in Figure 6 is encouraged.

**Response:** Thanks for the comments. Following the reviewer's suggestion, we have point out the uncertainty (~10%) of Mie calculation related to refractive index (RI). In this study, we used the RI values of coating materials and rBC cores were 1.5-0i and 2.26-1.26i, respectively, which were same with the ones used in our previous study (Zhang et al., 2018). In that work, we have estimated the uncertainty of ~10% from the assumptions of *RI* in Mie calculation. To make this point clear, we have added the related discussion in the revised manuscript, as *"The uncertainty of Mie calculation from the assumptions was estimated to be ~10% in our previous work (Zhang et al., 2018)."*

Correspondingly, we have added the uncertainty in the new Fig. 3 and Fig. 6 (Fig. R3 and Fig. R4 in the response). The related statements was also added in the captions.

[Figure]

Figure R3 (new Fig. 3b in the revised manuscript). The reduction in $m_{NR-CM}/m_{rBC}$ ratio of BC-containing particles during APEC relative to that before and after APEC. The error bar shown in (b) represents the uncertainties (~10%) of $m_{NR-CM}/m_{rBC}$ ratio from

Mie calculation.

[Figure]

Figure R4 (new Fig. 6b in the revised manuscript). The reduction in $E_{ab}$ of BC-containing particles during APEC relative to that before and after APEC. The error bar shown in (b) represents the uncertainties (~15%) of $E_{ab}$ from Mie calculation.

(3) Page 6 line 11: The authors attribute the decrease in ambient rBC and NO2 on the emission controls. It is encouraged to give more discussion show diurnal variations of their concentration (Sun, Y. et al. "APEC Blue": Secondary Aerosol Reductions from Emission Controls in Beijing. Sci. Rep. 6, 20668; doi: 10.1038/srep20668 (2016), as well as back trajectories of air mass during APEC and non-APEC period to support it.

**Response:** Thanks for the comments. Following the reviewer's suggestion, we have given more discussion showing diurnal variations of rBC and $NO_2$ concentration (Fig. R5 in the response and Fig. S4 in the supplement), as well as back trajectories of air mass during APEC and non-APEC period (Fig. R6 in the response and Fig. S3 in the supplement) to support the decrease in ambient rBC,

NO$_2$ due to the emission controls. The related discussion was added "*Figure S4 shows that the diurnal variations of the rBC, NO$_2$ and SO$_2$ concentration and the PBL during the pollution episodes before, during and after APEC. Comparing the diurnal variations between the rBC concentration and the PBL revealed that the rBC concentrations during the pollution episodes were dominated by the PBL. However, the precursor concentration of secondary aerosol (i.e., NO$_2$ and SO$_2$) during the pollution episodes exhibited different diurnal variations with a peak at noontime and early afternoon, which was most likely attributed to regional transport. The back-trajectory analysis (Fig. S3) revealed that the air mass during the pollution episodes was mainly from polluted regions (i.e., Hebei and Tianjin). This indicated that regional emission controls would reduce the pollutant (i.e., rBC, NO$_2$ and SO$_2$) concentration in Beijing under polluted conditions. The Sun et al., (2016) has demonstrated significant reductions in the precursors of secondary aerosol during APEC compared to those in non-APEC period due to emission controls during over a regional scale (i.e., Beijing and adjacent areas). The similar PBL (Fig S4) during the pollution episodes before, during and after APEC further identified the important contribution of emission reduction to the decrease of rBC, NO$_2$ and SO$_2$ concentration during APEC.*"

[Figure]

Figure R5 (Fig. S4 in the revised manuscript). Diurnal variations of the rBC, $NO_2$ and $SO_2$ concentration and the PBL for the pollution episodes before, during and after APEC.

[Figure]

Figure R6 (Fig. S3 in the revised manuscript). Spatial distribution (0.25°×0.25°) of the effective emission intensity (EEI, defined by Lu et al. (2012)) for BC transported to the observation site (40º00'17" N, 116º19'34" E) for the pollution episodes before, during and after APEC. The EEI calculation was stated in our previous study (Zhang et al. 2018).

(4) Page 6 line 24: It is hard to understand the meaning of the sentence "The similar mode of … and similar atmospheric processes (coagulation and wet removal) for rBC particles … APEC." The predominance of rBC in ~100 nm range is mostly due to incomplete combustion processes (vehicle engine etc.). It is better to remove it.

**Response:** Thanks. Following the reviewer's suggestion, we have removed it.

(5) Page 7 line 4: "condensational growth was more effect". Please provide more information to support such statement.

**Response:** Thanks to the reviewer for raising this concern. Sorry for inappropriate statement. We have revised the sentence as "*The size-dependent $m_{NR-CM}/m_{rBC}$ ratio of BC-containing particles revealed that particle growth was more effective for smaller particles, which followed the diffusion-controlled growth law (Seinfeld and Pandis 2006).*"

(6) In the third paragraph, Where is "Fig. 4b"? What is the reason for the difference in diurnal variability of Dc before and after APEC period?

**Response:** Thanks to the reviewer for raising these questions. Fig. 4b is described in the fourth paragraph, as "*However, the reduction in $m_{NR-CM}/m_{rBC}$ ratio of*

*BC-containing particles for the pollution episode during APEC compared to that after APEC showed a different diurnal cycle, with maxima at ~10:00-12:00 LT and with minima at ~15:00-17:00 LT (Fig. 4b)....*"

To the second question, sorry for the misleading and Figure 4(a) and (b) do not show the difference in diurnal variability of Dc before and after APEC period. Figure 4 shows the difference in diurnal variability of the reduction in $m_{NR-CM}/m_{rBC}$ ratio of BC-containing particles with size-resolved $D_c$ during APEC compared with those before (Fig. 4(a)) and after (Fig. 4(b)) APEC period. The difference was mostly likely to be attributed to different formation mechanism of BC coating materials before and after APEC. In the pollution episode before APEC, the production of BC coating materials was dominated by the photochemistry. However, the effect of other processes (e.g., heterogeneous chemistry) on the production of BC coating materials were enhanced during the pollution episode after APEC. We have discussed this point in the third and fourth paragraphs of page 7 in the manuscript.

**References:**

Lu, Z., Streets, D. G., Zhang, Q., and Wang, S.: A novel back-trajectory analysis of the origin of black carbon transported to the Himalayas and Tibetan Plateau during 1996–2010, *Geophys. Res. Lett.*, 39, 2012.

Seinfeld, J. H., and Pandis, S. N.: Atmospheric Chemistry and Physics: From Air Pollution to Climate Change, 2nd ed. *John Wiley & Sons, Inc.*, New York, 2006.

Sun, Y., Wang, Z., Wild, O., Xu, W., Chen, C., Fu, P., Du, W., Zhou, L., Zhang, Q., Han, T., Wang, Q., Pan, X., Zheng, H., Li, J., Guo, X., Liu, J., and Worsnop, D. R.: "APEC Blue": Secondary Aerosol Reductions from Emission Controls in Beijing, *Sci. Rep.*, 6, 20668, 2016.

Zhang, Y., Zhang, Q., Cheng, Y., Su, H., Li, H., Li, M., Zhang, X., Ding, A., and He, K.: Amplification of light absorption of black carbon associated with air pollution, *Atmos. Chem. Phys. Discuss.*, https://doi.org/10.5194/acp-2017-983, in review, 2018.

Zhang, Y., Zhang, Q., Cheng, Y., Su, H., Kecorius, S., Wang, Z., Wu, Z., Hu, M., Zhu, T., Wiedensohler, A., and He, K.: Measuring the morphology and density of internally mixed black carbon with SP2 and VTDMA: new insight into the absorption enhancement of black carbon in the atmosphere, *Atmos. Meas. Tech.*, 9, 1833-1843, 2016.

---

## Author Comment (AC3) · 27 Jun 2018

C. He

cenlinhe@ucar.edu

The authors combined SP2 measurements and Mie theory calculations to provide evidence for the reduction of black carbon (BC) light absorption due to the APEC emission control. The paper is well written. I have one minor suggestion on the uncertainty associated with the calculation/analysis of BC light absorption.

We would like to thank the reviewer for the valuable and constructive comments, which helps us to improve the manuscript. Listed below are our responses to the comments point-by-point, as well as the corresponding changes made to the revised manuscript. The reviewer's comments are marked in black and our answers are marked in blue, and the revision in the manuscript is further formatted as '*Italics*'.

Recent observations (e.g., China et al., 2015; Wang et al., 2017) have shown various complicated BC coating structures/morphology, which are not core-shell. Further modeling studies (e.g., Scarnato et al., 2013; He et al., 2015, 2016) have indicated a large variation in BC absorption and scattering due to the observed complex particle coating structures/morphology. Thus, assuming a core-shell structure in the present study may lead to uncertainty in the estimate of BC light absorption. It would be helpful if the authors could include these recent studies and add some discussions on this issue.

**Response:** According to the referee's suggestion, we have added the following discussion in the manuscript, as shown below:

*"The actual shape of BC-containing particles in the atmosphere was complex (China et al., 2015; He et al., 2015; He et al. 2016; Scarnato et al. 2013; Wang et al., 2017). In this study, we focused on investigating the BC-containing particles during pollution episodes. Under polluted conditions, we have found fully aged BC-containing particles in Beijing, China (Zhang et al., 2018). In our previous study (Zhang et al., 2016), we found that the thickly coated BC particles in the north china*

*plain (including Beijing) exhibited near-spherical shape and a core-shell structure used in the Mie calculation was reasonable."*

**References:**

China, S., et al.: Morphology and mixing state of aged soot particles at a remote marine free troposphere site: Implications for optical properties, *Geophys. Res. Lett.*, 42, 1243–1250, doi:10.1002/2014gl062404, 2015.

He, C., et al.: Variation of the radiative properties during black carbon aging: theoretical and experimental intercomparison, *Atmos. Chem. Phys.*, 15, 11967-11980, doi:10.5194/acp-15-11967-2015, 2015.

He, C., et al.: Intercomparison of the GOS approach, superposition T-matrix method, and laboratory measurements for black carbon optical properties during aging, *J. Quant. Spectrosc. Radiat. Transf.*, 184, 287–296, doi:10.1016/j.jqsrt.2016.08.004, 2016.

Scarnato, B. V., et al.: Effects of internal mixing and aggregate morphology on optical properties of black carbon using a discrete dipole approximation model, *Atmos. Chem. Phys.*, 13, 5089–5101, doi:10.5194/acp-13-5089-2013, 2013.

Wang, Y., et al.: Fractal dimensions and mixing structures of soot particles during atmospheric processing, *Environ. Sci. Technol. Lett.*, 4, 487-493, doi:10.1021/acs.estlett.7b00418, 2017.